# Assessing Stress Induced by Fluid Shifts and Reduced Cerebral Clearance during Robotic-Assisted Laparoscopic Radical Prostatectomy under Trendelenburg Positioning (UroTreND Study)

**DOI:** 10.3390/mps7020031

**Published:** 2024-04-01

**Authors:** Tobias Peschke, Matthias Feuerecker, Daniel Siegl, Nathalie Schicktanz, Christian Stief, Peter Zu Eulenburg, Alexander Choukér, Judith-Irina Buchheim

**Affiliations:** 1Laboratory of Translational Research “Stress and Immunity”, Department of Anesthesiology, LMU University Hospital, LMU Munich, 81377 Munich, Germany; tobias.peschke@med.uni-muenchen.de (T.P.); matthias.feuerecker@med.uni-muenchen.de (M.F.); alexander.chouker@med.uni-muenchen.de (A.C.); 2Department of Anesthesiology, LMU University Hospital, LMU Munich, 81377 Munich, Germany; daniel.siegl@med.uni-muenchen.de; 3Division of Cognitive Neuroscience, Faculty of Psychology, University of Basel, 4001 Basel, Switzerland; nathalie.schicktanz@unibas.ch; 4Department of Urology, LMU University Hospital, LMU Munich, 81377 Munich, Germany; christian.stief@med.uni-muenchen.de; 5Institute for Neuroradiology, LMU University Hospital, LMU Munich, 81377 Munich, Germany; peter.zu.eulenburg@med.uni-muenchen.de

**Keywords:** robotic-assisted laparoscopic radical prostatectomy, neurofilament light chain, neuronal damage, optic nerve sheet diameter, single-molecule array, neuromonitoring

## Abstract

In addition to general anesthesia and mechanical ventilation, robotic-assisted laparoscopic radical prostatectomy (RALP) necessitates maintaining a capnoperitoneum and placing the patient in a pronounced downward tilt (Trendelenburg position). While the effects of the resulting fluid shift on the cardiovascular system seem to be modest and well tolerated, the effects on the brain and the blood–brain barrier have not been thoroughly investigated. Previous studies indicated that select patients showed an increase in the optic nerve sheath diameter (ONSD), detected by ultrasound during RALP, which suggests an elevation in intracranial pressure. We hypothesize that the intraoperative fluid shift results in endothelial dysfunction and reduced cerebral clearance, potentially leading to transient neuronal damage. This prospective, monocentric, non-randomized, controlled clinical trial will compare RALP to conventional open radical prostatectomy (control group) in a total of 50 subjects. The primary endpoint will be the perioperative concentration of neurofilament light chain (NfL) in blood using single-molecule array (SiMoA) as a measure for neuronal damage. As secondary endpoints, various other markers for endothelial function, inflammation, and neuronal damage as well as the ONSD will be assessed. Perioperative stress will be evaluated by questionnaires and stress hormone levels in saliva samples. Furthermore, the subjects will participate in functional tests to evaluate neurocognitive function. Each subject will be followed up until discharge. Conclusion: This trial aims to expand current knowledge as well as to develop strategies for improved monitoring and higher safety of patients undergoing RALP. The trial was registered with the German Clinical Trials Register DRKS00031041 on 11 January 2023.

## 1. Introduction

A sudden change in body position leads to rapid fluid shifts that are well compensated by the cardiovascular system in combination with an acute catecholaminergic stress response. Prolonged fluid shifts, however, may not be completely compensated for (e.g., standing for several hours and subsequent peripheral edema). In addition to general anesthesia and mechanical ventilation, robotic-assisted laparoscopic radical prostatectomy (RALP) requires maintaining a capnoperitoneum and placing the patient in a steep downward tilt (Trendelenburg position). The effects of the resulting fluid shift on the cardiovascular system seem to be modest and well tolerated [1,2]. The intracerebral physiology under these conditions and how these conditions affect intracranial pressure (ICP) and cerebral perfusion pressure (CPP) is less clear.

In animal models, it has been shown that the invasively measured ICP can increase by up to 10 mmHg due to the capnoperitoneum and the downward tilt [3,4,5,6,7,8]. In humans, a non-invasive estimation of the intracranial pressure and the detection of increased intracranial pressure can be achieved with the help of ultrasound-guided measurement of the optic nerve sheath diameter (ONSD). A non-invasive method to estimate changes in the ICP in humans is to measure the optical nerve sheath diameter (ONSD) using ultrasound [9,10,11]. Several studies on patients undergoing RALP have shown that the ONSD can increase significantly [12,13,14]. A study from 2014 observed a 12.5% increase in the ONSD in a small subset of these patients (three out of twenty). ONSD values corresponding to an ICP of over 20 mmHg were observed in patients who showed valve insufficiency of the internal jugular vein (IJV) during RALP [14]. In addition, the choice of the anesthesia protocol seemed to affect the compensatory mechanisms. The intracranial autoregulation of the human body can maintain an adequate CCP in most situations. Only when these mechanisms are exhausted do severe complications occur [15,16]. In this context, it has been shown that using total intravenous anesthesia with propofol instead of employing an anesthetic gas (e.g., desflurane or sevoflurane) seemed to be beneficial [17,18].

The effect on postoperative cognitive performance in this patient population remains underinvestigated and is therefore not conclusively understood. Patients with a high intraoperative ONSD are more likely to appear disoriented during the awakening period or in the recovery room [19]. Due to the increasing use of RALP (80% of radical prostatectomies in the USA [20]), it is important to better investigate neurological complications and to develop a better monitoring strategy and awareness of potential transient brain damage. Some studies did not detect a cognitive deficit in patients after RALP [13]. Others, however, showed a lower outcome in standardized tests such as the mini-mental state examination (MMSE) in patients who experienced intraoperative IJV valve insufficiency [14]. Additionally, an increased risk of postoperative delirium was noted [21].

The correlation between postoperative (emergence) delirium and the observed intraoperative valve insufficiency suggests that an intraoperative fluid shift might occur leading to an endothelial dysfunction and reduced cerebral clearance, possibly resulting in transient neuronal damage. A very specific biomarker for neuronal damage that is increasingly gaining importance is the axonal protein neurofilament light chain (NfL) [22]. Previous work showed that NfL might serve as a prognostic marker not only in neuronal disease states but also in several other patient groups, e.g., after cardiac arrest [23,24] and cardiac surgery [25,26].

Research on astronauts, previously conducted by our team, showed that long-term space missions and exposure to microgravity may induce a cephalad fluid shift resulting in an elevation of biomarkers for traumatic brain injury and neurodegeneration in blood [27]. Using the highly sensitive single-molecule array (SiMoA) before and after a long-term space mission (mean duration 169 days), a significant increase in NfL and glial fibrillary acidic protein (GFAP, a biomarker for neurotrauma) was detected. This was likely due to the long-lasting pressure caused by the fluid shift in microgravity and the restitution of the blood–brain barrier after return to Earth. These changes correlated with volumetric brain tissue changes observed in magnetic resonance images (MRI) in the very same group [28,29]. A long-term space mission is significantly different from RALP and cannot be compared. However, the data showed that even in healthy individuals blood-based biomarkers for brain damage may serve as a monitoring tool for transient brain damage due to cephalad fluid shifts.

We hypothesize that the pronounced downward tilt in RALP in combination with the capnoperitoneum and general anesthesia may restrict physiological adaptation mechanisms resulting in reduced endothelial integrity and a fluid shift into the tissue. These processes might lead to a detectable increase of blood biomarkers for brain damage and changes in the ONSD. The prospective UroTreND Study aims to create a better understanding of cerebral and hormonal autoregulation and pathomechanisms in patients undergoing RALP compared to patients who are scheduled for the conventional open technique. Furthermore, the study shall support the development of strategies for improved monitoring tools such as the ONSD and thus further the safety of patients undergoing RALP.

## 2. Outcome Measures

### 2.1. Primary Endpoint

The primary endpoint of this study is the perioperative concentration of NfL in plasma (single-molecule array (SiMoA)).

### 2.2. Secondary Endpoints

Secondary endpoints are as follows:The markers of endothelial and blood-brain barrier function in the blood (matrix metalloproteinases-2 and -9, zonulin, S100 calcium-binding protein B (S100B), neuron-specific enolase (NSE), GFAP, amyloid-β proteins 40 and 42 (Aβ40, Aβ42), and glycocalyx proteins (e.g., hyaluronan, syndecan-1, and heparan sulfate));The estimation of ICP through measuring the ONSD using ultrasound and a short screening of extracranial cervical blood vessels for valve insufficiencies;Neurocognitive function (MMSE, clock drawing test, Nu-DESC, memory test);The physiological parameters of stress (cortisol in saliva and urine, current stress test (CST));The presence and severity of headache;Immune analyses (differential, leukocyte subsets, plasma cytokines, cell adhesion molecules, markers of cellular activation, radical production, phagocytosis);The analysis of immune cell signaling cascades using transcriptional analysis (mRNA) and Western blot (whole blood, PBMCs, isolated neutrophils);Procedure-related parameters (end-tidal CO_2_, type of anesthesia protocol, use of anesthetic gas, blood loss, mean arterial pressure, incision–suture time, total time of anesthesia);An MRI scan of the cerebrum (optional).

The detection method for each endpoint is listed under “4. Analysis methods”.

## 3. Trial Design

The UroTreND Study is a single-center, interventional, non-randomized, prospective clinical trial. It will be conducted at the LMU University Hospital Munich, Germany. The trial was registered under the German Clinical Trial Register (Trial number: DRKS00031041).

### 3.1. Recruitment

Potential subjects will be identified at the outpatient clinic of the urology or anesthesiology department based on their diagnosis or scheduled surgery. Due to the investigated procedure, only male subjects will be eligible for recruitment. Only patients presenting with organ-centered tumors who are scheduled for a radical prostatectomy either in the robotic-assisted or in the conventional open technique will be recruited. In most cases, patients will choose their preferred method of surgery. No science team member will influence the choice of the type of procedure and no randomization will be performed. Each patient will receive all standard preoperative examinations (e.g., cardiac and pulmonary auscultation). We will screen eligible patients for study participation according to the selection criteria. Information regarding the trial and study-specific procedures will be provided during an interview with a member of the study team. Informed consent will be obtained from all subjects. In addition to the informed consent to trial participation, the patient will be asked to provide a separate signature for the MRI scan. The latter consent is independent of participation in the study. The subject will be informed that the MRI scan is optional. We will inform all subjects who receive an MRI about scheduling, risks, and benefits, especially the risk of incidental findings in the brain. Patients will receive an adequate amount of time to think about their choice (in most cases, weeks) before they sign a written informed consent to participate in this study.

### 3.2. Timeline for Each Patient

After informed consent to participate in the study has been given, the baseline time point T0 will take place. On the day of the surgery, there are three time points scheduled: T1 after the induction of anesthesia, T2 1–2 h after incision, and T3 during emergence, after the patient has been repositioned. The follow-up period will consist of T4 on postoperative day one, T5 on postoperative day three, and T6 before discharge (usually postoperative days 5–7). An overview of the patients’ schedule of activities according to the standard protocol items—recommendations for interventional trials (SPIRIT)—is provided in Figure 1.

### 3.3. Anesthesiologic Management and Postoperative Care

All patients will receive anesthesiologic management according to the institution’s standards. This involves the placement of two peripheral venous lines and general anesthesia. Neuroaxial procedures will not be used. Anesthesia will be induced with propofol (2 mg/kg), sufentanil (0.4 mg/kg), and rocuronium (0.6 mg/kg), and maintained with propofol and remifentanil or sevoflurane in patients with certain conditions (e.g., cardiopulmonary diseases). Pulmonary ventilation will be performed according to the needs of the patient and clinical standards. The responsible anesthesiologist will decide about the nature and extent of additional monitoring measures and the anesthesia protocol in both study groups. Participants of the study team will not influence the choice of anesthesia protocol or any other clinical decisions about the treatment of both patient groups.

### 3.4. Trial Population and Selection Criteria

The study population will consist of patients who undergo an elective radical prostatectomy (scheduled for RALP or conventional open technique).

#### 3.4.1. Inclusion Criteria

Subjects must meet the following inclusion criteria to be eligible for enrolment:Able to legally state written informed consent;Age: >18 years and <80 years;Planned radical prostatectomy.

#### 3.4.2. Exclusion Criteria

Subjects showing the following exclusion criteria cannot be included in the trial:Not able to give informed consent;Missing written informed consent;Known neurological disease with or without increased intracranial pressure;Known psychiatric disease with or without permanent medication;Known eye disease (e.g., glaucoma);Known severe lung disease;Immunosuppressive medication;Participation in a research project/clinical trial that conflicts with this study;Known severe autoimmune disease (ASA group III or higher);Known alcohol, drug, or medication abuse.

### 3.5. Sample Size Estimation

The experimental group (n = 30) will receive RALP while the control group (n = 20) will receive a conventional open radical prostatectomy. Thus, a total of 50 patients will be included. Each patient will choose the preferred method for himself. To obtain homogenous groups, attention will be paid to a balance of the preoperative risk assessment (ASA score) in both groups.

We hypothesize that the postoperative NfL concentration will be higher in the experimental group compared to the control group. To date, NfL has not been evaluated in this patient population. However, it is expected that the increase in the NfL concentration will be minor in the control group compared to the experimental group. Although the temporal dynamics are currently unknown, we expect an increase between the preoperative measurement point and the postoperative measurement points until discharge (usually days 5–7).

In a recent study comparing patients after bypass surgery with patients after neck surgery, an increase in concentration of NfL was found on the seventh postoperative day using SiMoA. The sample size of this study group was 25 and the control group was 26 [30]. In patients undergoing neck surgery, no NfL was detected. This trial will also employ SiMoA. RALP may be different from bypass surgery; however, a risk of neuronal damage can be assumed, and previous work showed an increased ICP in a similar group strength [12,13]. We therefore estimated the required sample size in the experimental group to be 30 patients. It may, however, be possible that NfL is detectable in the control group and increases over time but to a lesser extent or in a narrower time window. Thus, we expect a smaller dispersion of detectable changes in the control group. Everything considered, we calculated the control group to be up to 20 patients.

### 3.6. Statistical Evaluation

The exact case number can only be approximated due to the lack of previous studies in this patient population. The obtained data from each patient will be analyzed individually. After testing for normal distribution (Kolmogorov–Smirnov test), the appropriate statistical test for comparison will be selected (e.g., repeated measurements ANOVA or Friedman test). The median, mean values, and standard deviations will then be calculated from the generated data sets of all patients. The data will be statistically analyzed using the software SPSS 24.0 (IBM, New York, NY, USA) and Sigma Plot 13 (Chicago, IL, USA). A significance level α of 0.05 and a statistical power of 0.8 were defined for this study. A flow chart of the trial is provided in Figure 2.

### 3.7. Ethics and Good Clinical Practice

The trial will be conducted in accordance with the Clinical Trials Directive 2001/20/EC of the European Parliament and the Council of the EU, the International Conference on Harmonization guidance regarding Good Clinical Practice (ICH-GCP E6 R1), the relevant national regulations, and the Declaration of Helsinki. Ethical approval was obtained from the local Ethics Committee of the LMU Medical Faculty, Munich, Germany (protocol code 22-0562; 20 October 2022). Any modifications to the protocol will be immediately communicated to all responsible authorities. A completed SPIRIT checklist was provided as a Appendix A of this manuscript (Appendix A).

### 3.8. Risk–Benefit Assessment

The study will require several study-specific measures on the patient. Some of them will be clinically necessary and most of the study-related measures will be non-invasive. The only invasive procedure will be the blood draw, where 20 mL will be drawn from each patient at seven time points (maximum total 140 mL). These blood draws will be carried out via existing catheters or as part of routine blood sampling where possible.

Each patient might already benefit directly through the monitoring of cognitive function. In both groups, occurring cognitive impairments (e.g., due to the choice of anesthetics) might be detected early and will be well documented. This will also lead to the introduction of countermeasures, which should, however, not affect the overall outcome of the study. Additionally, a better understanding of the adaptation in this particular position and a gain in knowledge regarding neuronal damage will be possible for the experimental group. This might contribute to an improvement in patient monitoring and safety. The easy-to-perform and non-invasive control of the ONSD [31] might be established as a standard monitoring tool within the department of anesthesiology and thus further improve the training of young colleagues.

### 3.9. Data Management and Privacy

All data input, processing, and analysis will be carried out in accordance with the Data Protection Act. Only employees of the UroTreND Study will have access to all study data, and they are obliged to confidentiality. The data will be protected from unauthorized access. All patient-related data will be recorded in pseudonymized form (letter and number code): After signing the informed consent, each patient will be randomly assigned to an alphanumeric number of a defined number group. These alphanumeric numbers will then be encrypted together with the personal data.

Samples that are processed for this study will only contain the alphanumeric number and therefore will not provide any personal reference. A patient identification list including the full name, date of birth, and alphanumeric number will be maintained by the employees of this study, stored safely, and remain protected. This list will be necessary for decryption in case of medical or scientific reasons. Additional data from routine examinations or measurements will be added to the research database according to the alphanumeric pseudonymization. The patient identification list will be stored for up to 10 years after the conclusion of the trial. Each patient will be thoroughly informed about data management and privacy.

Patients will have the right to withdraw their consent and cancel their study participation at any time without being required to give any reasons. In such cases, the patient will be asked if they agree to the further use of their data/samples in an anonymized form, or if they prefer their data/samples to be deleted or destroyed.

### 3.10. Quality Control and Assurance

The Department of Anesthesiology of the LMU University Hospital is certified according to the highest national standard qualification system that is also recognized globally (DIN-ISO-9001). This includes the quality management of clinical trials. All employees are specifically trained according to their tasks (e.g., data management and privacy, handling of measurement instruments). Simple plausibility checks are automatically carried out when entering new data. Furthermore, checks on the completeness of the data are regularly performed. This serves to identify and rectify quality defects at an early stage.

## 4. Analysis Methods

Blood samples will be taken at all time points T0–T6. The ONSD will be measured at T0–T4. Functional tests and questionnaires will be conducted at T0/T1 and T3–6 simultaneously for saliva collection. Urine samples will be taken, when possible, at T1 and T3/T4. The MRI scan will be performed at T0 and T4. An overview of the samples and procedures employed at each time point is provided in Figure 3.

### 4.1. Blood Processing

The concentration of NfL, as well as the concentrations of NSE, GFAP, Aβ40, and Aβ42, will be quantified by the highly sensitive SiMoA in EDTA plasma [32]. Matrix metalloproteinases-2 and -9, zonulin, S100B, hyaluronan, syndecan-1, and heparan sulfate will be measured with ELISA. Standard immune analyses such as differentials will be conducted at the Department of Clinical Chemistry at the LMU Hospital Munich. Functional analyses, leukocyte subsets, plasma cytokines, cell adhesion molecules, and markers of cellular activation will be evaluated using flow cytometry and multiplex assays on a Luminex^®^ xMAP^®^ platform (Merck Millipore, Darmstadt, Germany). The expression of signaling cascades of interest will be assessed facultatively via transcriptome (mRNA) and Western blot analyses. No genomic research (e.g., chromatin structure, mutations, single-nucleotide polymorphism) will be performed.

### 4.2. ONSD/Extracranial Blood Vessels

Focused sonography will be performed according to standardized operating procedures [31]: The optic nerve will be marked at a depth of 3 mm behind the retinal plane. At this depth, the transverse diameter will then be determined by a pair of measuring points, which are placed directly lateral to the reflex-rich cerebrospinal fluid space. As measurement inaccuracies cannot be ruled out, the measurement will be repeated three times, and the median value will be used. This procedure will be completed in under two minutes to avoid heat-induced eye damage [33]. In addition to the measurement of the ONSD, a screening of the extracranial cervical blood vessels will be performed (visual impairment of valve functionalities, flow changes).

### 4.3. Neurocognitive Tests

The functional tests are non-invasive and standardized.

MMSE: The MMSE is an established screening instrument for testing spatial orientation, memory, attention, numeracy, and language. To test orientation, ten questions are asked about time and place.Clock drawing test: The patient will be asked to write the numbers 1–12 and the time 11:10 in a provided circle on a sheet of paper. This very simple test is used to assess instruction comprehension, execution planning, and working memory [34]. It will be carried out in combination with the MMSE.The Nursing Delirium Screening Scale (NU-DESC) is an established screening test for delirium evaluation in under 3 min [35].Memory test: This test will be computer-based. The patient will be asked to memorize and reproduce a series of words (episodic memory) and numbers (digit span test). Additionally, the attention and reaction time is tested by clicking on a red button (red button task).

The test will contain the following parts:Learning of words with immediate recall;Red button task;Digit span forwards;Digit span backwards;Recall of words learned during part 1.

### 4.4. Questionnaires

Two types of questionnaires will be used in this study. The first one will be the standardized and validated current stress test (CST), (Appendix A). It consists of six polar adjective pairs that describe the current well-being on a six-point rating scale [36]. The second one will be an evaluation form where the recruited subject can report the presence and graded severity of headaches.

### 4.5. Saliva Sampling and Processing

Saliva will be collected via a salivette (Sarstedt, Nümbrecht, Germany) in a fasting state, according to the manufacturer’s instructions. Additionally, saliva will be collected via passive flow into another tube. The samples will be stored at −20 °C until analysis. Cortisol will be measured with ELISA.

### 4.6. Urine Sampling and Processing

The urine will be taken perioperatively from the collected urine in the indwelling catheter. Cortisol will be measured via high-performance liquid chromatography (HPLC) or ELISA.

### 4.7. MRI Scan

It is expected that an MRI scan will be performed in selected patients (minimum of n = 10 per group) before surgery and on postoperative day 1 to investigate the presence of morphological changes. MRI scans will be performed without contrast medium, employing a 3 Tesla device.

## 5. Discussion

Less invasive robotic-assisted surgery techniques are increasingly popular with both physicians and patients. While the benefits such as non-inferior surgical outcomes, faster recovery, cosmetic advantages, or economic benefits [37] may favor the implementation as a standard, the requirements to enable access to the site of surgery and the associated risks need to be better understood. RALP requires the head-down tilt positioning of the patient, which is a challenge for anesthesiologists. Patients with planned RALP often only undergo screening tests for prevalent cardiopulmonary and endocrine diseases [15]. Yet, no screening procedure for a prevalent, yet unknown, neurological disease or vulnerability to neuronal damage is performed as standard. Moreover, research on the anesthesiologic management of patients undergoing RALP is limited. Although brain tissue oxygenation seems not to be affected even after three to four hours of steep Trendelenburg positioning [38,39], several studies reported a rise in the ONSD [13,19] and intraocular pressure [40]. We believe that more research is warranted to investigate the effect of steep head-down tilt on brain tissue homeostasis. Recent advances in biomarker research have shown that NfL might serve as a good candidate for monitoring potential transient brain damage in these patients [23,24]. Moreover, establishing the ONSD as a non-invasive standard tool to monitor patients as well as a better selection and preoperative evaluation of patients might improve patient safety and clinical practice.

### 5.1. Limitations

To our knowledge, this is the first time that blood-based biomarkers will be analyzed in this specific patient group. To investigate the current clinical standard, we purposely did not want to change the way patients are selected for the procedure and did not want to introduce a preselection bias by expanding the preoperative screening tools to detect a previously unknown neurological disease in the patient. Although patient safety would be increased, it is currently not conducted in the clinical setting. The neuro-cognitive screening tests that will be used in this study are well-established bedside tests that might be used in the future. The best-suited monitoring tool remains to be established, and the UroTreND Study aims to explore ways of implementing such screening procedures. Through the use of the ONSD and blood-based biomarkers, we hope to contribute to better patient care by identifying patients at risk.

In this pilot study, we will not be able to study the long-term effects of potential neurological damage due to the short postoperative observation period. Moreover, different anesthesiologic management strategies cannot be compared. These will be the focus of future studies. This study aims to gain first insights into whether the implementation of non-invasive monitoring tools and improved patient selection combined with blood-based biomarkers could enhance patient safety and clinical practice in RALP.

### 5.2. Conclusions

With this comprehensive study set-up, we aim to expand our knowledge of neurological vulnerability and damage in patients undergoing RALP as well as develop strategies for improved monitoring and safety of patients at risk.

Trial status: In preparation.

The trial is in the final stages of preparation. Recruitment shall commence in March 2024. This manuscript is based on the study protocol Version 1.2_31.08.2023.

## Figures and Tables

**Figure 1 mps-07-00031-f001:**
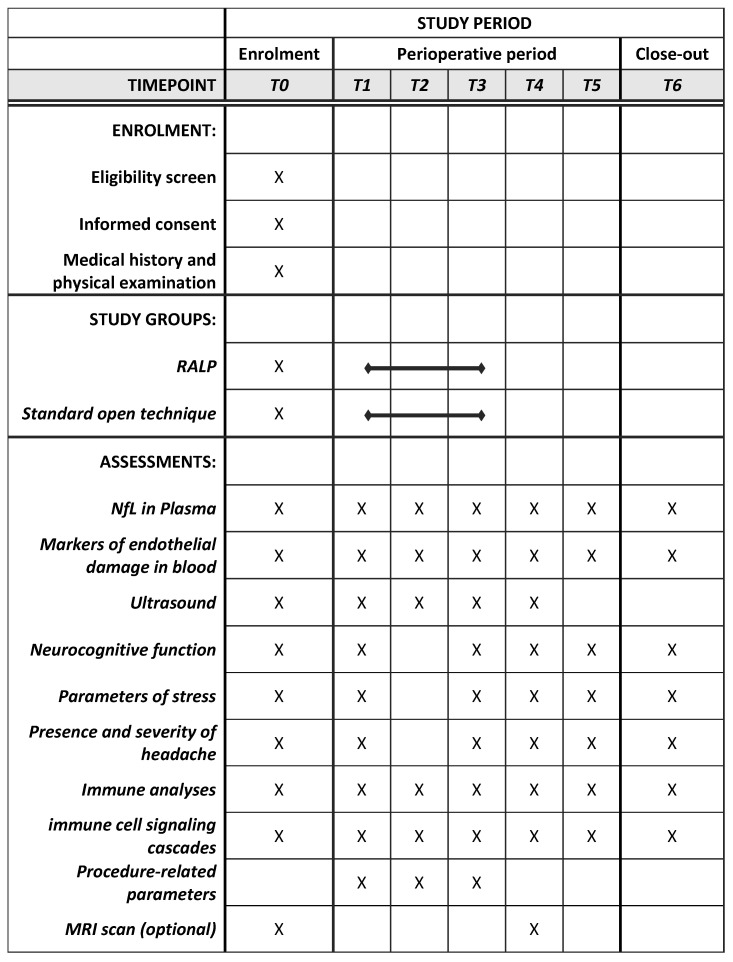
Overview of patients’ schedule of activities.

**Figure 2 mps-07-00031-f002:**
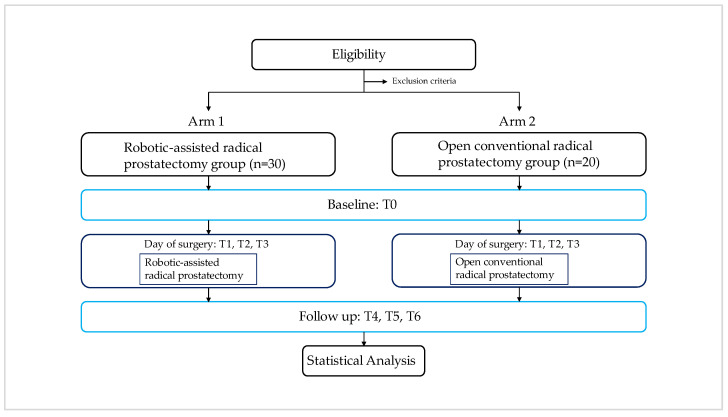
A flow chart diagram of the UroTreND Study.

**Figure 3 mps-07-00031-f003:**
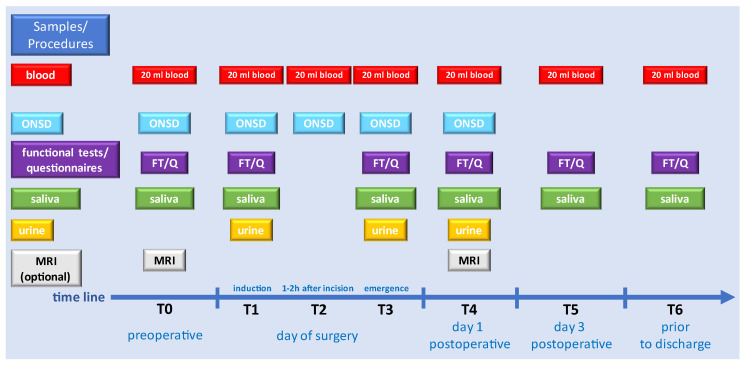
Timeline of samples and procedures for each patient (ONSD: optical nerve sheet diameter; MRI: magnetic resonance imaging).

## Data Availability

Access to the personal data of participants as well as to the final data set will only be granted to designated trial investigators and team members handling the primary data of this trial. Data confidentiality will be respected at all times. The results of this trial will be published in an international open-access journal so that they are publicly available. The original dataset that will be collected in the time course of the study will be made available without identifiable data of subjects upon reasonable request directed to the corresponding author.

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
