# Peer review of "Assessing Stress Induced by Fluid Shifts and Reduced Cerebral Clearance during Robotic-Assisted Laparoscopic Radical Prostatectomy under Trendelenburg Positioning (UroTreND Study)"

_mps, 2024, doi:10.3390/mps7020031_

Round 1

Reviewer 1 Report

Comments and Suggestions for Authors

Dear authors, I would like to share my opinion after reading your paper.

The manuscript presents a comprehensively detailed research protocol for the UroTreND Study, which thoughtfully investigates the impact of fluid shifts and reduced cerebral clearance during robotic-assisted laparoscopic radical prostatectomy (RALP) in the Trendelenburg position. The study commendably focuses on understanding the effects on the brain and the blood-brain barrier, insightfully considering the unique challenges posed by the steep downward tilt required in RALP. The primary endpoint, the perioperative concentration of neurofilament light chain (NfL) in blood, is a well-chosen marker for neuronal damage. Admirably, the trial also seeks to compare RALP with conventional open radical prostatectomy across a significant cohort of 50 subjects. The inclusion of secondary endpoints, such as markers for endothelial function, inflammation, neuronal damage, and the optic nerve sheath diameter (ONSD), enriches the study's scope. Additionally, the assessment of perioperative stress through questionnaires, stress hormone levels in saliva samples, and functional tests for neurocognitive function are notably valuable aspects of this research. This study undeniably contributes significantly to our understanding of the neurological implications of RALP and has the potential to substantially improve patient safety and monitoring strategies.

Regarding the limitations, which I humbly suggest considering in the context of this otherwise robust study, I would like to point out the following aspects for potential reflection and further elaboration:

1.       Consideration for Neurological Screening: While patients planned for RALP typically undergo comprehensive screenings for prevalent cardiopulmonary and endocrine diseases, incorporating a standard procedure for screening unknown neurological diseases or vulnerability to neuronal damage could potentially enhance the study's applicability and patient safety.

2.       Expansion of Research on Anesthesiologic Management: Given the challenges of the steep Trendelenburg positioning required in RALP, a more detailed exploration in the realm of anesthesiologic management could provide invaluable insights and contribute to the broader body of knowledge in this field.

3.       Further Research on Brain Tissue Homeostasis: While existing studies suggest brain tissue oxygenation remains unaffected in the steep Trendelenburg position, a more in-depth investigation into the long-term effects of this position on brain tissue homeostasis would be a commendable addition to this study.

4.       Exploration of Transient Brain Damage Monitoring: The potential of neurofilament light chain (NfL) as a marker for monitoring transient brain damage is an exciting development in biomarker research. Further exploration and validation in this area could significantly advance our understanding and application of these biomarkers.

5.       Implementation of Non-Invasive Monitoring Tools: The establishment of optic nerve sheath diameter (ONSD) as a non-invasive standard tool for patient monitoring, coupled with improved patient selection and preoperative evaluation, could serve as a pivotal step in enhancing patient safety and clinical practice in RALP.

In closing, these thoughtful considerations underscore the need for more focused research in certain areas to further enhance the safety and efficacy of robotic-assisted laparoscopic radical prostatectomy procedures. I kindly request the esteemed authors to consider reflecting on these points in the limitations section or provide their valuable insights on the observations mentioned above.

Reviewer 2 Report

Comments and Suggestions for Authors

1. How to measure primary endpoint and secondary endpoints? Please add them.

2. Please rewrite 3.6. Statistical estimation. The authors should indicate the statistical tests, explain whether the data is normally distributed and which test was employed to identify the data distribution. Then, show the software, p value, a statistical power.

3. Please add references in sections 3.8-3.10, 4.1-4.2

4. The authors should be consistent about the tenses used to write the manuscript.

5. Please add more discussion on your results.

6. Please check grammar with language editor.

Comments on the Quality of English Language

For a scientific report, the clean and direct English is preferred.  May I suggest to ask for an English-native editor to help smoothing the whole paper?

Reviewer 3 Report

Comments and Suggestions for Authors

It is a valuable study protocol and wish you success in completing the study. 

Reviewer 4 Report

Comments and Suggestions for Authors

COMMENTS TO  mps-2827250

The article studies the possible brain involvement due to fluid pressure in the forced Trendeleburg position, for a long time, in robotic radical prostatectomy.

A defect that the authors must correct is that the abstract and the article lack conclusions.

They should add that section.
